# CardAIc-Agents: A Multimodal Framework with Hierarchical Adaptation for Cardiac Care Support

**Yuting Zhang**[1]      YTZ300@STUDENT.BHAM.AC.UK
[1] *School of Computer Science, University of Birmingham, UK*

**Karina V. Bunting**[2]      K.V.BUNTING@BHAM.AC.UK

**Asgher Champsi**[2]      A.CHAMPSI@BHAM.AC.UK
[2] *Department of Cardiovascular Sciences, University of Birmingham, UK*

**Xiaoxia Wang**[2,3]      XIAOXIA.WANG@UHB.NHS.UK
[3] *NIHR Birmingham Biomedical Research Centre and West Midlands NHS Secure Data Environment, University Hospitals Birmingham NHS Foundation Trust, UK*

**Wenqi Lu**[4]      W.LU@MMU.AC.UK
[4] *Department of Computing and Mathematics, Manchester Metropolitan University, UK*

**Alexander Thorley**[1]      AJT973@STUDENT.BHAM.AC.UK

**Sandeep S Hothi**[5]      S.HOTHI@NHS.NET
[5] *Department of Cardiology, Heart and Lung Centre, Royal Wolverhampton NHS Trust, UK*

**Zhaowen Qiu**[6]      249600398@QQ.COM
[6] *College of Computer and Control Engineering, Northeast Forestry University, China*

**Baturalp Buyukates**[*1]      B.BUYUKATES@BHAM.AC.UK

**Dipak Kotecha**[*2,3,7]      D.KOTECHA@BHAM.AC.UK
[7] *Julius Center, University Medical Center Utrecht, the Netherlands*

**Jinming Duan**[*1,8]      J.DUAN@BHAM.AC.UK
[8] *Division of Informatics, Imaging and Data Sciences, University of Manchester, UK*

**Editors:** Accepted for publication at MIDL 2026

## Abstract

Cardiovascular diseases (CVDs) remain the foremost cause of mortality worldwide, a burden worsened by a severe deficit of healthcare workers. Artificial intelligence (AI) agents have shown potential to alleviate this gap through automated detection and proactive screening, yet their clinical application remains limited by: 1) rigid sequential workflows, whereas clinical care often requires adaptive reasoning that selects specific tests and, based on their results, guides personalised next steps; 2) reliance solely on intrinsic model capabilities to perform role assignment without domain-specific tool support; 3) general and static knowledge bases without continuous learning capability; and 4) fixed unimodal or bimodal inputs and lack of on-demand visual outputs when clinicians require visual clarification. In response, a multimodal framework, CardAIc-Agents, is proposed to augment models with external tools and adaptively support diverse cardiac tasks. First, a CardiacRAG agent generates task-aware plans from updatable cardiac knowledge, while the Chief agent integrates tools to autonomously execute these plans and deliver decisions. Second, to enable adaptive and case-specific customization, a stepwise update strategy is developed to dynamically refine plans based on preceding execution results, once the task is assessed as complex. Third, a multidisciplinary discussion team is proposed which is automatically invoked to interpret challenging cases, thereby supporting further adaptation. In addition, visual review panels are provided to assist validation when clinicians raise concerns.

Zhang Bunting Champsi Wang Lu Thorley Hothi Qiu Buyukates Kotecha Duan

Experiments across three datasets showed the efficiency of CardAIc-Agents compared to mainstream Vision–Language Models (VLMs) and state-of-the-art agentic systems. Code will be publicly available at https://github.com/ytz300/CardAIc-Agents.

**Keywords:** Multimodal framework, medical AI agents, workflow optimization, cardiac applications, foundation models, echocardiographic imaging

## 1. Introduction

Cardiovascular diseases (CVDs) are the leading cause of mortality worldwide, accounting for 17.9 million deaths each year (Almeida et al., 2024). Notably, up to 80% of these deaths occur in low- and middle-income countries, where specialised care is limited (Bulto and Hendriks, 2024), which, combined with a global shortage of over 4 million healthcare workers (Vedanthan and Fuster, 2011), drives the demand for scalable and accessible cardiovascular care solutions. Recent advancements in large language models (LLMs) have led to human-level performance on challenging tasks; for instance, Med-PaLM has outperformed clinicians on the United States Medical Licensing Examination (Singhal et al., 2025). Despite these achievements, however, clinical practice, particularly for complex chronic conditions (e.g., heart failure (HF)), often relies on multimodal data for diagnosis, prognosis, and treatment (Weintraub, 2019). This gap underscores the need for multimodal strategies that extend beyond language-only models to more effectively support clinical practice.

While vision-language models (VLMs) such as LLaVA-Med (Li et al., 2023) and MedGemma (Sellergren et al., 2025) have fueled anticipation for medical multimodal artificial intelligence (AI), several challenges remain. For example, they are restricted to static images, whereas dynamic inputs such as echocardiograms are vital for cardiac function assessment. In addition, such generalist models retain static knowledge, which hinders their ability to assimilate evolving medical evidence. While Retrieve Augmented Generation (RAG) mitigates this challenge to some extent, traditional retrieval methods still present notable limitations. For example, Term Frequency Inverse Document Frequency (TF-IDF) relies on lexical matching but is limited in semantic comprehension, while Dense Passage Retrieval (DPR) encodes queries and documents into embeddings for similarity based retrieval yet often lacks semantic relevance (Karpukhin et al., 2020; Mallen et al., 2023).

Crucially, complex cardiovascular management often requires multi-step reasoning and a coordinated sequence of clinical actions, rather than a single-step response (McDonagh et al., 2021). Although prompt engineering techniques such as Chain-of-Thought (CoT) (Wei et al., 2022) partially mitigate this limitation by decomposing problems into substeps, model performance remains constrained by their intrinsic capabilities. The recent introduction of function calling and the Model Context Protocol (MCP) (Hou et al., 2025) provides a complementary pathway, enabling models to integrate external tools automatically and access standardized functions. These advances drive the development of AI agents capable of reasoning, planning, memory utilization, and action execution (Chang et al., 2024). However, most existing VLM-based agents in medicine still rely on assigning roles to models with static and generic knowledge, and often lack cross-turn memory, limiting their suitability for real-world cases that require multidisciplinary deliberation.

Another limitation of these existing VLM-based agents lies in their rigid and sequential workflows (Kim et al., 2024). Although recent advances have enabled ReAct-based

frameworks (Yao et al., 2023) to perform stepwise reasoning with intermediate outcomes and external tools, these frameworks still lack global planning capability. In contrast, MedAgent-Pro (Wang et al., 2025) offers disease-level planning, but its plans are generated before receiving patient-specific input, which may be effective for some routine tasks yet risks misalignment with individual clinical contexts. For example, echocardiogram view identification typically follows a fixed workflow (e.g., commercial software, manual view selection), whereas complex HF diagnosis involves diverse patient presentations that require tailored test orders (e.g., ECG, echocardiography) and subsequent personalised management based on results. These observations highlight the need for flexible frameworks capable of both task-level and case-level adaptation across diverse clinical contexts. In addition, current VLM agents lack intermediate visual outputs, such as the left ventricular contour delineations, which are critical for clinical verification in complex or uncertain cases.

Motivated by the above, in this study, an adaptive framework, CardAIc-Agents (comprising CardiacRAG and a Chief agent), is introduced to augment models with external tools, enabling autonomous execution of cardiac tasks (e.g., diagnosis, echocardiogram view extraction, segmentation, detection of P, QRS, and T waves) across diverse modalities (e.g., textual, signal, image, and video). Specifically, the CardiacRAG agent is developed to formulate general plans based on the latest domain knowledge and the proposed hybrid retrieval technique, whereas the Chief agent enhances its own capabilities through the integration and orchestration of external tools for plan execution and definitive decision-making. To support adaptive planning across tasks and patient-specific cases, the system initially assesses task complexity, executes the plan, and dynamically refines it as new evidence emerges. For more challenging cases, a multidisciplinary discussion team (MDT), augmented with external tools and cross-turn memory, is proposed to support further interpretation. Finally, when clinicians raise concerns, visual review panels are provided for validation. In summary, the key contributions can be articulated as follows:

- A domain-specific framework, CardAIc-Agents, is developed to enhance the capabilities of large models through specialized tool integration, enabling autonomous execution of diverse cardiac tasks on multimodal data.

- Adaptive strategies are proposed to stratify task complexity, refine plans iteratively as new evidence emerges, initiate team discussions, and provide visual validation, enabling hierarchical adaptation tailored to specific tasks and individual patients.

- A CardiacRAG agent is introduced to derive plans based on an updated cardiac knowledge base, while employing a hybrid retrieval mechanism to optimize semantic comprehension and relevance.

- A multidisciplinary discussion team is designed to integrate external tools that extend the static and general capabilities of foundation models and to incorporate cross-turn memory that preserves context across reasoning steps.

## 2. Method

CardAIc-Agents consist of two components: the CardiacRAG and the Chief agent (Figure 1). The former, based on a dedicated cardiac knowledge base, generates and updates

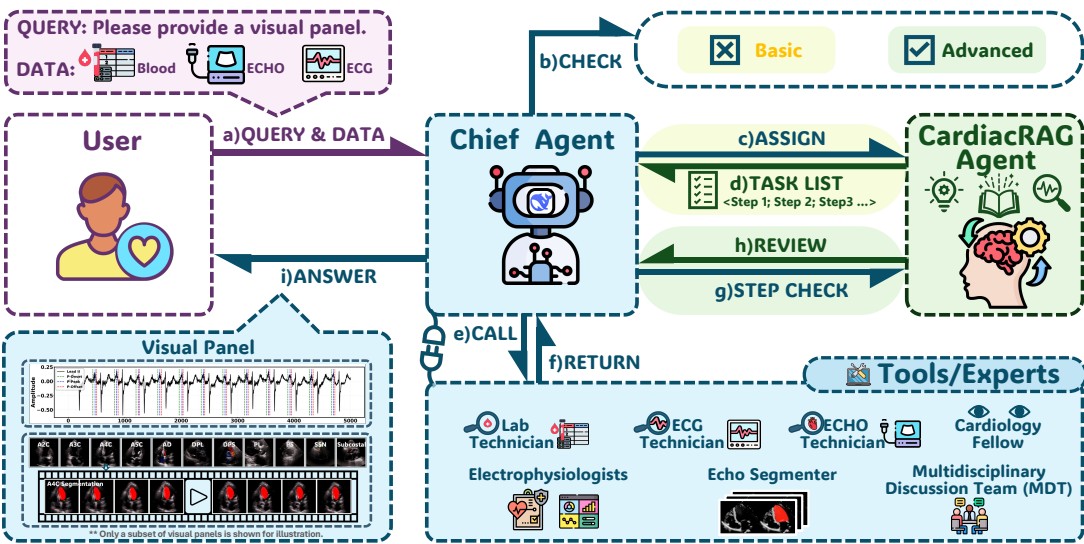

Figure 1: Overview of the CardAIc-Agents framework. The workflow proceeds from (a) to (i). For tasks identified as basic, steps (g) and (h) are skipped; for advanced tasks, the full pipeline is executed.

plans as new evidence emerges. The Chief agent serves as the primary decision-maker, responsible for complexity assessment, task assignment, plan execution, and tool invocation. Together, these components support *hierarchical adaptation* in CardAIc-Agents, consistent with clinical workflows. Specifically, *hierarchical adaptation* denotes 1) task-complexity stratification (Basic vs. Advanced), 2) iterative plan refinement as new evidence is incorporated, 3) automatic on-demand initiation of the multidisciplinary discussion team (MDT), and 4) optional visual outputs for further validation.

As shown in Figure 1, upon receiving the query with associated multimodal data such as ECGs (a), the Chief agent performs a complexity assessment (b) and assigns the task to the CardiacRAG agent (c), which retrieves domain-specific evidence from a curated cardiac knowledge base and constructs a general plan for the case (d). For low-complexity cases, the Chief agent executes this plan by invoking the required analytical tools (e), integrates their results (f), and generates the final clinical response (i). For high-complexity cases, the plan is adaptively revised (g, h) as new evidence is incorporated, enabling iterative refinement of subsequent reasoning and tool selection before the final decision is produced. For more challenging cases, the Chief agent may autonomously initiate the MDT to provide enhanced interpretation. When clinicians raise concerns, visual review panels may be required as an available option for human validation.

## 2.1. CardiacRAG Agent

The CardiacRAG agent is developed as current LLMs and VLMs encode static knowledge and their general-purpose design may lack cardiovascular domain specificity. This agent emulates the clinician reasoning process through information retrieval from authoritative medical sources and it is structured into three key stages (see Figure 2).

*Knowledge base construction.* To reduce the complexity of information retrieval and improve accuracy, the knowledge base construction process focuses exclusively on cardiac content. This domain-specific approach selectively aggregates data $\{D_i\}_{i=1}^M = \{D_1, D_2, \ldots, D_M\}$ from authoritative medical sources, including major US academic medical centers (e.g., Mayo Clinic 2025), UK National Health Service (2025), health information platforms (e.g., MedlinePlus 2000), and recently published official guidelines (see Appendix A for details).

Then, the raw documents $\{D_i\}_{i=1}^M$ are preprocessed through a transformation function $T$ that extracts and normalizes textual content, producing clean text:

$$\{S_i\}_{i=1}^M = T\big(\{D_i\}_{i=1}^M\big), \quad i = 1, 2, \ldots, M. \tag{1}$$

where $T$ refers to BeautifulSoup (Abodayeh et al., 2023) for HTML files and Docling (Livathinos et al., 2025) for PDFs, and $M$ refers to the total number of collected documents.

Finally, cleaned texts $S_i$ are split into chunks $s_i^j$ to preserve contextual continuity:

$$s_i^j = \text{chunk}_j(S_i; d_s, d_o), \quad j = 1, 2, \ldots, L_i, \tag{2}$$

where $L_i$ is the chunk count of document $i$; $j$ is the chunk index; and $d_s$ and $d_o$ denote the chunk and overlap size, respectively.

*Hybrid retrieval.* To reduce irrelevant results from current vector similarity retrieval techniques, a hybrid retrieval method, combined with TF-IDF variants, is applied to further filter results using domain-specific keywords and ensure clinical specificity (see Figure 2).

• Vector similarity retrieval. To preserve semantic relevance, both chunks $s_i^j$ and the query are embedded via Bio_ClinicalBERT (Alsentzer et al., 2019) as $v_i^j = \phi(s_i^j)$ and $q = \phi(\text{query})$. The set of document vectors $\mathcal{V} = \bigcup_{i=1}^M \{v_i^j\}_{j=1}^{L_i}$ is ranked by cosine similarity:

$$\text{sim}(q, v_i^j) = \frac{q \cdot v_i^j}{\|q\|\|v_i^j\|}, \tag{3}$$

and the top $3n$ vectors $(\mathcal{V}_{(1)}, \ldots, \mathcal{V}_{(3n)})$ are returned, where $n$ is the final number of results. Document vectors are indexed and stored in the FAISS vector database for efficient retrieval and reused without recalculation in subsequent queries.

• Keyword-based filtering. To improve clinical relevance, top $3n$ documents are further filtered based on domain-specific weights:

$$MW(k) = \begin{cases} \omega_{\text{medical}}[k], & k \in \mathcal{B}_{\text{medical}} \\ 1, & \text{otherwise} \end{cases}, \tag{4}$$

where $k$ is the keyword from the query $Q$, $\mathcal{B}_{\text{medical}}$ is the clinical vocabulary, and $\omega_{\text{medical}}$ is the term importance defined based on the clinical context. To exploit structural cues, a position-based bonus (Hofstätter et al., 2021) is introduced:

$$PB(k, s_i^j) = \begin{cases} 1.2, & \text{if } \text{pos}(k, s_i^j) < 0.3 \times d_s \\ 1, & \text{otherwise} \end{cases}, \tag{5}$$

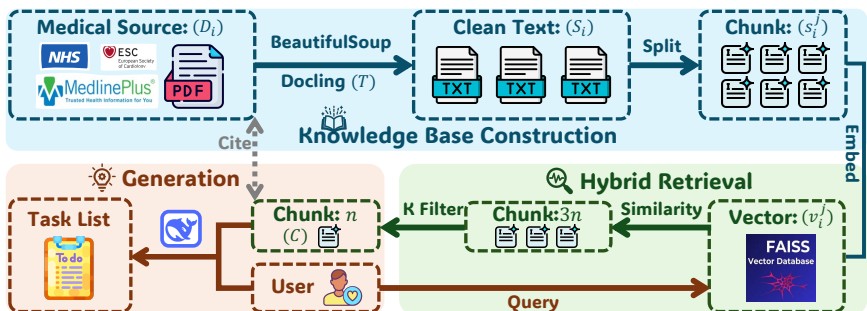

Figure 2: Illustration of the CardiacRAG agent. $D_i$ denotes the $i$-th source, $S_i$ is the cleaned text, $s_i^j$ is the $j$-th chunk from source $i$, $v_i^j$ is its corresponding vector, $T$ represents the transformation method, $K$ denotes keyword-based filtering, $n$ is the number of chunks retrieved, $c$ is the final retrieved content, and *Cite* indicates optional return of original chunks for transparency and reference.

where $\text{pos}(k, s_i^j)$ is the first index of $k$ in chunk $s_i^j$. The final retrieval score then is:

$$\text{Score}_{(s_i^j, Q)} = \frac{1}{|Q|} \sum_{k \in Q} TF(k, s_i^j) \cdot MW(k) \cdot PB(k, s_i^j), \tag{6}$$

where chunks with scores above threshold $\theta$ are retained, and term frequency defined as:

$$TF(k, s_i^j) = \frac{\text{count}(k, s_i^j)}{|\text{words}(s_i^j)|}. \tag{7}$$

*Guideline generation.* Based on the retrieved chunks $C = \{c_1, c_2, \ldots, c_n\}$ and the query $Q$, the general plan is generated by the CardiacRAG agent, which employs DeepSeek-R1-Distill-Qwen-32B (DeepSeek-AI, 2025) as its core model. This general plan could be continuously updated as new results become available during subsequent steps, thereby reflecting clinical practices.

• General plan. Given the query $Q$, the model does not reconstruct the knowledge base; instead, it retrieves relevant evidence from the prebuilt cardiac knowledge base (FAISS vector database described earlier) using the hybrid retrieval mechanism. The retrieved chunks $C$, together with the query, provide the contextual input from which the model generates an initial stepwise plan (see Appendix D.3 for the prompt):

$$P = \texttt{DeepSeek-R1}(\texttt{PlanPrompt}(Q, C)), \tag{8}$$

where the plan $P = (p_1, \ldots, p_s)$ contains $s$ steps, which may vary by case.

• Stepwise update. At each step, the CardiacRAG agent evaluates the execution state and determines whether the next action should be revised based on intermediate evidence. Formally, at step $i$, the model receives the execution history $\log_i$ and the proposed next step $p_{i+1}$, and produces an updated decision:

$$(S_i, A_i, p'_{i+1}) = \texttt{DeepSeek-R1}\big(\texttt{UpdatePrompt}(\log_i, p_{i+1})\big), \qquad i = 1, \ldots, s-1. \tag{9}$$

Here, $S_i$ summarizes the evidence, $A_i \in \{\text{stop}, \text{continue}\}$ specifies whether execution terminates, and $p'_{i+1}$ is the updated next step; if no revision is required, $p'_{i+1} = p_{i+1}$. The procedure halts at the first index $k$ such that $A_k = \text{stop}$, or proceeds through all $s$ steps otherwise. A detailed case study is provided in Appendix E.3.

## 2.2. Chief Agent

The Chief agent leverages the advanced reasoning capabilities of DeepSeek-R1 to coordinate specialized tools and apply adaptive strategies that adjust behaviour at both the task and patient levels, reflecting clinical workflows and supporting real-world application.

*Adaptive strategies.* Given the query $Q$, the Chief agent first assesses the task complexity:

$$\ell = \texttt{DeepSeek\_R1}(\texttt{Prompt}(Q)), \qquad \ell \in \{\text{basic}, \text{advanced}\}. \tag{10}$$

Based on the predicted complexity level $\ell$, the agent executes either the general plan or a stepwise refinement procedure. Execution in both modes follows:

$$p^*_{i+1} = \begin{cases} p_{i+1}, & \ell = \text{basic}, \\ p'_{i+1}, & \ell = \text{advanced}, \end{cases} \qquad i = 1, \ldots, k. \tag{11}$$

At each step, the Chief automatically executes the actual action $p^*_{i+1}$ by invoking tool $T_{i+1} \in \mathcal{T}$ (see Appendix B for the used tools), yielding the output $t_{i+1} = T_{i+1}(p^*_{i+1})$. For complex cases, the Chief may additionally invoke the MDT tool to simulate clinical case conferences. The evidence is then appended to the log via $\log_{i+1} = \texttt{Append}(\log_i, t_{i+1})$. After all steps, the Chief synthesizes the log to generate the final summary and answer:

$$(\text{Summary}, \text{Answer}) = \texttt{DeepSeek-R1}(\log). \tag{12}$$

Further, the overall system can provide visual validation when required for disputed or ambiguous cases (see Figure 4), allowing clinicians to perform manual verification:

$$V = \texttt{CardAIc-Agents}(Q), \tag{13}$$

which encapsulates the above workflow before producing the final visual output $V$.

*Multidisciplinary discussion team (MDT).* As shown in Figure 3, this team reviews inputs and intermediate outputs from tools to support comprehensive decisions. First, the Chief designates two relevant domain-expert roles based on the inputs. Each expert independently analyzes the inputs, and their respective results are synthesized by the Chief. The two experts then review this synthesis together with outputs produced by tools, enabling the integration of information sources that extend beyond the large model review paradigm typically adopted in existing medical agents. During this review, each expert provides an explicit binary agreement signal (agree/disagree) to indicate whether their current judgment is concordant with the synthesized conclusion from the preceding step. The Chief subsequently re-synthesizes the updated information, completing one discussion round. All intermediate results generated during each round are stored as memory for downstream use.

If consensus is achieved by both experts, or the maximum number of predefined discussion rounds is exhausted, the Chief issues the final decision. Otherwise, the synthesized

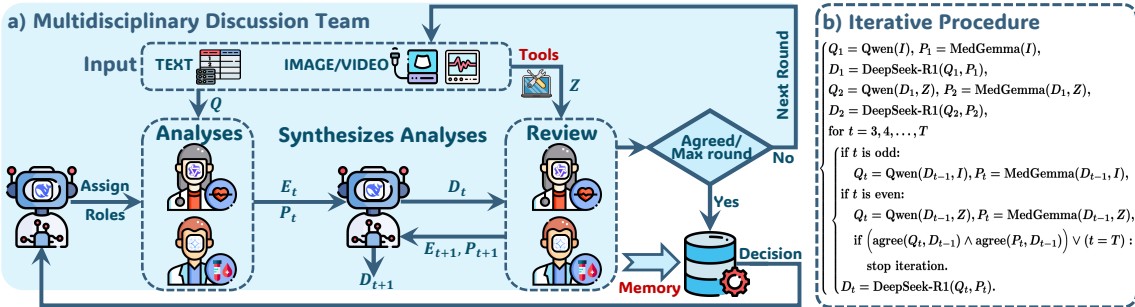

Figure 3: Multidisciplinary Discussion Team (MDT) workflow: an iterative loop of (i) Assign Roles, (ii) Analyses, (iii) Synthesizes Analyses, and (iv) Review, terminated by a decision from the Chief Agent. (b) Iterative inference (corresponding to (a)): DeepSeek-R1 performs (i) and (iii) and outputs the decision; Qwen and MedGemma handle (ii) and (iv) as experts. $Q$ denotes the original input, $Z$ the intermediate tool outputs, $E_*, P_*, D_*$ the model responses, with $T$ as the total steps and $t$ the step index.

output from the current round is combined with the original inputs and passed to the next round, ensuring that subsequent iterations built upon accumulated evidence rather than relying solely on the initial inputs or on the stochastic behavior of responses from LLMs or VLMs. In this study, the two experts are implemented using MedGemma (Sellergren et al., 2025), specialized for medical image analysis, and Qwen2.5-VL (Bai et al., 2023), specialized for video processing as noted in Figure 3(b).

## 3. Experiments

### 3.1. Experimental Settings

*Datasets.* CardAIc-Agents is evaluated on three datasets: (i) MIMIC-IV (Johnson et al., 2023), for HF diagnosis, which includes data from 1,524 patients with three modalities (laboratory test results, 12-lead ECGs, and echocardiograms (ECHOs)); (ii) PTB-XL (Wagner et al., 2020), for Myocardial infarction (MI) diagnosis (Strodthoff et al., 2023), which includes data from 10,147 patients with structured patient information, ECG-derived variables, and 12-lead ECGs; (iii) The PTB Diagnostic ECG Database contains patient information and 12-lead ECGs from 268 cases for HF prediction (see Appendix C for details).

*Metrics and baseline.* The diagnostic performance is evaluated using the area under the receiver operating characteristic curve (AUC) and accuracy (Yu et al., 2021), with 95% confidence intervals. For the visual outputs, two cardiologists independently assess and score the results. The proposed agent is compared with medical VLMs, including LLaVA-Med (Li et al., 2023) and MedGemma (Sellergren et al., 2025), with MedGemma combined with CoT (Wei et al., 2022) and ReAct (Yao et al., 2023) to evaluate step-by-step reasoning and tool-augmented strategies. Comparisons are also made with medical agent frameworks such as MedAgents (Tang et al., 2024), ReConcile (Chen et al., 2024), and MDAgents (Kim et al., 2024). Further implementation details are provided in Appendices D.1 for the proposed method and D.4 for the baseline agent, respectively.

Table 1: Performance Comparison Across Methods and Datasets

| Category | Method | MIMIC-IV | | | PTB-XL | | PTB Diagnostic | |
|---|---|---|---|---|---|---|---|---|
| | | ACC | AUC | Latency | ACC | AUC | ACC | AUC |
| **VLMs & Variants** | LLaVA-Med (Li et al.) | 0.35(0.30,0.41) | 0.34(0.28,0.40) | 22.515 | 0.39(0.37,0.42) | 0.51(0.47,0.55) | 0.12(0.08,0.16) | 0.44(0.34,0.59) |
| | MedGemma (Sellergren et al.) | 0.76(0.71,0.80) | 0.82(0.77,0.86) | 15.046 | 0.56(0.53,0.59) | 0.55(0.52,0.59) | 0.76(0.71,0.81) | 0.88(0.83,0.93) |
| | MedGemma (CoT,Wei et al.) | 0.65(0.60,0.70) | 0.81(0.76,0.86) | 359.038 | 0.53(0.50,0.56) | 0.54(0.50,0.58) | 0.58(0.52,0.64) | 0.75(0.64,0.88) |
| | MedGemma (ReAct,Yao et al.) | 0.67(0.62,0.72) | 0.71(0.66,0.76) | 139.422 | 0.83(0.81,0.85) | 0.83(0.80,0.85) | 0.69(0.64,0.75) | 0.72(0.41,0.99) |
| **Medical Agents** | MedAgents (Tang et al.) | 0.74(0.70,0.79) | 0.82(0.78,0.87) | 156.870 | 0.65(0.62,0.68) | 0.62(0.58,0.66) | 0.75(0.70,0.79) | 0.84(0.74,0.92) |
| | ReConcile (Chen et al.) | 0.49(0.44,0.55) | 0.75(0.69,0.80) | 103.206 | 0.43(0.40,0.46) | 0.57(0.53,0.61) | 0.55(0.49,0.61) | 0.76(0.58,0.93) |
| | MDAgents (Kim et al.) | 0.52(0.47,0.58) | 0.61(0.55,0.68) | 64.563 | 0.56(0.52,0.58) | 0.60(0.56,0.64) | 0.74(0.68,0.79) | 0.74(0.53,0.96) |
| **Fine tuned VLMs** | Qwen2.5-VL (Bai et al.) | 0.78(0.73,0.82) | 0.85(0.81,0.90) | **1.173** | 0.93(0.91,0.94) | 0.96(0.94,0.97) | 0.72(0.66,0.77) | 0.80(0.61,0.99) |
| | Janus-Pro (Chen et al.) | 0.84(0.79,0.88) | **0.91(0.88,0.94)** | 1.247 | **0.96(0.95,0.97)** | **0.99(0.98,0.99)** | 0.75(0.70,0.80) | 0.83(0.57,0.99) |
| **Proposed** | CardAIc-Agents | **0.87(0.82,0.90)** | 0.89(0.85,0.93) | 79.137 | **0.96(0.95,0.97)** | 0.96(0.94,0.98) | **0.77(0.72,0.82)** | **0.88(0.65,1.00)** |

Note: Boldface values indicate best performance within each dataset and metric; Values in parentheses represent 95% confidence intervals.
Latency is reported per sample; ACC = accuracy; AUC = Area Under the Curve; CoT = Chain of Thought; ReAct = Reasoning and Acting.

## 3.2. Results and Comparative Analysis

*Comparison with VLMs and variants.* As shown in Table 1, CardAIc-Agents outperformed all baseline VLMs across the three cardiac datasets. The largest gap was observed on the MIMIC-IV, where CardAIc-Agents achieved an accuracy of 0.87 compared to only 0.35 by LLaVA-Med ($p < 0.05$). This was partly due to the limited token input length, which constrained the performance of this medical yet general VLM. Secondly, MedGemma performed the best among the VLMs, while enabling CoT reasoning did not improve performance across all datasets (see Appendix E.2 for confusion metric analysis). Finally, its ReAct system, built on LangChain for tool use, improved PTB-XL performance but not MIMIC-IV, and still was outperformed by the proposed method.

*Comparison with medical agents.* CardAIc-Agents also outperformed state-of-the-art medical agents as shown in Table 1. Among these, ReConcile showed the largest gap, with accuracies of 0.49 (vs. 0.87) on MIMIC-IV, 0.43 (vs. 0.96) on PTB-XL, and 0.55 (vs. 0.77) on PTB Diagnostic ($p < 0.05$). A key limitation of these agents lay in their reliance on the intrinsic capabilities of models; inference remained constrained despite guidance from expert-role prompts. In addition, their static knowledge bases and rigid reasoning pipelines limited adaptation to diverse cases. By contrast, the proposed agent leveraged an updatable CardiacRAG agent, integrated external tools to augment model capabilities, and incorporated an adaptive strategy that enabled refinement and optimization across diverse cases.

*Comparison with fine-tuned VLMs.* The proposed agent achieved performance comparable to that of fine-tuned VLMs specifically optimized for their respective tasks. The results showed that it outperformed Qwen2.5-VL and achieved performance comparable to Janus-Pro on the MIMIC-IV and PTB-XL datasets. To assess generalization, all fine-tuned models were directly evaluated on the PTB Diagnostic Database for HF diagnosis without any task-

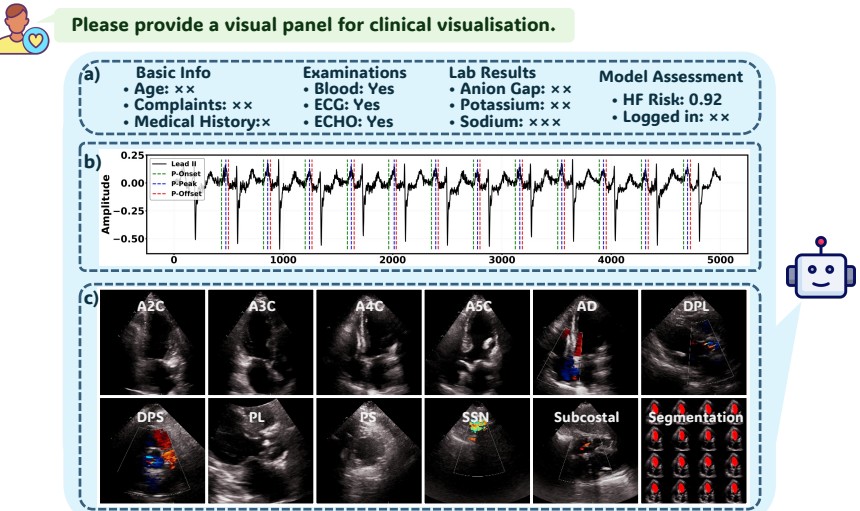

Figure 4: Visual panel generated by CardAIc-Agents: a) patient profile display b) ECG waveform with labeled P and T waves c) echocardiographic view identification with A4C segmentation video frames. Additional details are in the Appendices.

specific adaptation (Table 1). The results showed that Qwen2.5-VL and Janus-Pro achieved lower accuracies of 0.72 and 0.75, respectively, compared with 0.77 ($p < 0.05$).

Another finding was that the fine-tuned Janus-Pro achieved a higher AUC than the proposed agent on the first two datasets, but not higher accuracy. This difference might have stemmed from the tendency of LLM-based agents to assign elevated baseline probabilities even in the absence of explicit diagnoses, whereas fine-tuned models typically generated more precise probability estimates, often benefiting from higher numerical precision (e.g., 16- or 32-bit). Even so, results in Table 1 demonstrated that the proposed CardAIc-Agents achieved comparable, if not better, performance than VLMs specifically fine-tuned for a given task. Note that beyond generalization and task specificity, flexibility and interpretability are also concerns for fine-tuned models, whereas CardAIc-Agents does not require task-specific fine-tuning and serves as a general-purpose framework.

*Assessment of intermediate visual outputs.* CardAIc-Agents could provide on-demand support to clinicians for the validation of complex or uncertain cases, a capability enabled by the review panel introduced to facilitate this process (see Figure 4). This function was evaluated by two cardiologists. For echocardiography, the agent automatically identified 11 standard views from raw DICOM, achieving 100% accuracy in key views (eg., A3C, A4C, PLAX, PSAX, SC) and over 80% accuracy in others (a random sample of 10 cases); Left ventricle segmentation on A4C views had been reported in prior studies with a Dice Coefficient of 0.922 on the EchoNet-Dynamic dataset. Detection of P, QRS, and T waves from 12-lead ECGs was rated suboptimal by experts, mainly due to stringent criteria requiring precise identification of every heartbeat, indicating an area for further improvement.

Table 2: Ablation Study of Adaptive Workflow, CardiacRAG, and Tools.

| Level | Workflow | CardiacRAG | Tools | | | | ACC | AUC |
|---|---|---|---|---|---|---|---|---|
| | | | **EPhys** | **EchoSeg** | **CardioF** | **MDT** | | |
| Module | | ✓ | ✓ | ✓ | ✓ | ✓ | 0.80(0.75,0.85) | 0.87(0.83,0.91) |
| | ✓ | | ✓ | ✓ | ✓ | ✓ | 0.77(0.72,0.82) | 0.81(0.76,0.86) |
| | ✓ | ✓ | | ✓ | ✓ | ✓ | 0.84(0.80,0.88) | 0.88(0.83,0.92) |
| | ✓ | ✓ | ✓ | | ✓ | ✓ | 0.83(0.78,0.87) | 0.88(0.84,0.92) |
| | ✓ | ✓ | ✓ | ✓ | | ✓ | 0.77(0.72,0.81) | 0.84(0.79,0.88) |
| | ✓ | ✓ | ✓ | ✓ | ✓ | | 0.84(0.80,0.88) | 0.88(0.84,0.92) |

| Level | Workflow | Similarity | Filter | EPhys | EchoSeg | CardioF | Tool Memory | | ACC | AUC |
|---|---|---|---|---|---|---|---|---|---|---|
| Intra module | ✓ | | ✓ | ✓ | ✓ | ✓ | ✓ | ✓ | 0.83(0.79,0.87) | 0.87(0.83,0.91) |
| | ✓ | ✓ | | ✓ | ✓ | ✓ | ✓ | ✓ | 0.83(0.78,0.87) | 0.86(0.81,0.90) |
| | ✓ | ✓ | ✓ | ✓ | ✓ | ✓ | | ✓ | 0.75(0.70,0.80) | 0.83(0.78,0.87) |
| | ✓ | ✓ | ✓ | ✓ | ✓ | ✓ | ✓ | | 0.82(0.77,0.86) | 0.86(0.81,0.90) |
| Proposed | ✓ | ✓ | ✓ | ✓ | ✓ | ✓ | ✓ | ✓ | **0.87(0.82,0.90)** | **0.89(0.85,0.93)** |

✓ = enabled; empty = disabled; Boldface = best performance; MDT = multidisciplinary discussion team.
EPhys = electrophysiologists; EchoSeg = echo segmente; CardioF = cardiology fellow.

### 3.3. Ablation Studies

*Adaptive workflow.* The ablation study was conducted on the MIMIC-IV dataset to evaluate the contribution of the adaptive workflow, where accuracy improved from 0.80 to 0.87 ($p < 0.05$, Table 2). This result confirmed the effectiveness of reasoning in an incremental and feedback-aware manner. Specifically, the model performed step-by-step evaluation and summarization, allowing it to re-assess the current state at each stage and adjust the plan accordingly before proceeding. This process emphasized a global plan followed by stepwise adjustments, distinguishing it from a purely ReAct-based approach that prioritizes stepwise changes, as well as from strategies that rely solely on general planning. More detailed sensitivity analysis of task complexity assessment is provided in Appendix E.5.

*CardiacRAG agent.* The contribution of the CardiacRAG agent is shown in Table 2. The results indicated a clear improvement when a dedicated and independent agent was assigned to generate and refine plans based on domain knowledge, yielding a 10% performance improvement. This highlighted the effectiveness of the proposed module in precisely retrieving relevant information, as well as the valuable contribution of its curated domain-specific knowledge base. Furthermore, the intra-module ablation of the hybrid retrieval mechanism confirmed that using only vector similarity retrieval or only keyword-based filtering led to decreased performance (see Appendices E.1 and E.4 for parameter analysis and retrieved knowledge quality, respectively.).

*Multidisciplinary discussion team.* Table 2 also reports an increase in accuracy from 0.84 to 0.87 attributed to the proposed team ($p < 0.05$). This gain reflected two drivers: first, the effectiveness of this team to incorporate diverse multimodal information and assign distinct roles to specialized models for collaborative discussion; second, the capability of the agent to dynamically activate the tool upon detecting uncertainty or insufficient evidence in earlier reasoning stages, selectively engaging the team as needed. Such improvement also confirmed the benefits of the proposed adaptation strategy. In addition, intra-module ablation showed

Table 3: Inference Cost and Performance with On demand MDT.

| Method | Overall (Full Set) | | | | | Subset (Triggered Only) | | | | |
|---|---|---|---|---|---|---|---|---|---|---|
| | Latency | | Performance | | | | Latency | | Performance | |
| | Mean(s) | P95(s) | ACC | AUC | Trig_Rate(%) | | Mean(s) | P95(s) | ACC | AUC |
| No MDT | 50.44 | 62.98 | 0.84(0.80,0.88) | 0.88(0.84,0.92) | 0 | | 52.72 | 59.81 | 0.74(0.53,0.89) | 0.55(0.18,0.89) |
| On-demand MDT | 79.14 | 161.01 | 0.87(0.82,0.90) | 0.89(0.85,0.93) | 6 | | 194.31 | 285.25 | 0.84(0.68,1.00) | 0.61(0.23,1.00) |

Note: Latency is reported in seconds (s) per sample; Mean and P95 denote the average and 95th percentile latency, respectively;
Trig_Rate = trigger rate; ACC = accuracy; AUC = area under the curve; Values in parentheses indicate 95% confidence intervals.

that both the tool and its persistent memory contributed to performance, highlighting their importance for stable multi-agent coordination.

Notably, MDT was triggered for only 6% of samples, yet the mean latency increased from 50.44 s to 79.14 s, indicating a latency overhead associated with its invocation (see Table 3). Further analysis on the triggered subset showed that the per-case latency was 194.31 s, substantially higher than without MDT (52.72 s), while larger accuracy gains were achieved (ACC 0.84 vs. 0.74). This highlighted that on-demand MDT yielded performance improvements at the cost of latency overhead on a small subset of samples, which may constrain deployment. However, this trade-off parallels clinical escalation workflows, in which multidisciplinary discussion is reserved for a minority of complex cases and incurs substantial time overhead.

*Domain-specific tools.* Table 2 further quantifies the effect of each domain-specific tool. The results indicate that removing any single tool yields a consistent performance degradation, with the largest decrement observed when the Cardiology Fellow tool is ablated (ACC 0.87 vs. 0.77, $p < 0.05$). Note that the remaining tools (Laboratory Technician, ECG Technician, and Echocardiographer) are not ablated, as they are required for preprocessing the laboratory test results, 12-lead ECGs, and echocardiograms, respectively. In addition, the adaptive test-time scaling (TTS) mechanism is designed for procedural robustness (see Appendix D.2), while the final decision relies on multiple cross-modal evidence sources rather than any single external tool output, which limits the influence of isolated tool errors.

## 4. Conclusion

This study introduces CardAIc-Agents, a multimodal framework with adaptive capabilities for cardiac-related tasks. Experiments on three public datasets showed that it outperformed general medical VLMs and state-of-the-art medical agents. In summary, by combining external tools with a cardiac knowledge base, this study presents a hierarchical adaptive framework spanning: complexity assessment; iterative plan refinement as new evidence emerges; dynamic activation of specialized team discussions for complex cases; and provision of visual outputs to support clinician verification. With this adaptive design, CardAIc-Agents delivers scalable multimodal decision support and shows potential for deployment, particularly in resource-limited clinical settings.

## Acknowledgements

The research was conducted using the Baskerville Tier 2 HPC service, which was funded by the EPSRC and UKRI through the World Class Labs scheme (EP/T022221/1) and the Digital Research Infrastructure programme (EP/W032244/1). Baskerville is operated by Advanced Research Computing at the University of Birmingham.

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

## Appendix A. Knowledge Base Construction

To streamline the information retrieval and improve clinical accuracy, the knowledge base is curated with a targeted focus on cardiac-related content. It selectively integrates information from authoritative medical sources and the most recent official clinical guidelines to ensure domain relevance and content reliability. In addition, this base is intended to mitigate the limitations of static large language models by facilitating the timely integration of updated medical knowledge without requiring repeated fine-tuning of the core model.

### A.1. Authoritative Medical Sources (Web-based Resources).

Medical content is collected from leading academic medical centers, national health service, and reputable online health information platforms. These sources include:

- Mayo Clinic (2025). A widely recognized medical information platform developed by Mayo Clinic and its affiliates, offering clinically validated content for public health education. In our knowledge base, we primarily focus on its cardiovascular section, which provides structured disease overviews, diagnostic pathways, and treatment guidelines aligned with clinical best practices.

- Cleveland Clinic (2025). As a leading academic medical center with a strong emphasis on innovation and translational research, Cleveland Clinic provides comprehensive information on advanced diagnostic techniques, treatment strategies, and access to cutting-edge clinical trials, which are often not available through general platforms. Given these strengths, its heart disease–related content is incorporated into our cardiac knowledge base.

- UK National Health Service (2025). The NHS offers comprehensive and authoritative information on symptoms, conditions, treatments, risk factors, and self-management strategies. It also provides structured guidance for navigating healthcare services, making informed decisions, and responding to public health events. To enrich our knowledge base with UK-specific clinical pathways and patient-facing recommendations, we incorporate its cardiac-related web content.

- MedlinePlus (2000). MedlinePlus is a comprehensive online health information resource designed for patients and the general public. It is managed by the U.S. National Library of Medicine (NLM), which is part of the National Institutes of Health (NIH). As a leading American platform, it provides evidence-based, patient-centered information covering a wide range of health topics. To incorporate authoritative U.S.-focused cardiovascular content such as disease conditions, treatment options, and preventive care, materials from MedlinePlus are integrated into our knowledge base.

Note that other authoritative organizations, such as Healthline (LLC, 2025) and the National Heart, Lung, and Blood Institute (NHLBI) 2025, are also included. Institutions like the American Heart Association (AHA) (AHA, 2025) can provide valuable information; however, due to copyright and usage restrictions, their content is not directly incorporated into our knowledge base.

### A.2. Official Clinical Guidelines (Local Knowledge Repository).

The most recent authoritative guidelines from leading cardiology organizations are incorporated, including the latest versions of the European Society of Cardiology (ESC) guidelines (McDonagh et al., 2021), American Heart Association (AHA) guidelines (Perman et al., 2024), Canadian Cardiovascular Society (CCS) guidelines (Guerra et al., 2024), Chinese Society of Cardiology guidelines (Zhang et al., 2025b), among others. This local repository offers two key advantages: first, it enables direct compilation of the most relevant and well-controlled up-to-date content, providing the model with authoritative references to support

clinical decision-making; second, it circumvents issues related to automated data extraction from certain websites, such as the AHA, which restricts or limits web crawling. It is worth mentioning that all online data retrieval and access in this study are conducted strictly for research purposes only.

## Appendix B. Tools

CardAIc-Agents leverage a variety of tools that are not driven by large models, but are instead toolbox-style modules designed to perform specific domain functions. Each tool acts as an expert in its respective domain, collaboratively supporting the execution of cardiac tasks. The following are descriptions of these tools:

*Laboratory technician.* This tool preprocesses laboratory test results (Labs, tabular data) for downstream analysis by extracting clinical information such as demographics, laboratory values, and medication history from structured or semi-structured text, producing both natural language outputs and their tokenized representations:

$$(Lab_{\text{text}}, Lab_{\text{token}}) = \text{LabProcessor(Labs)}.$$

*ECG technician.* This tool preprocesses raw 12-lead ECGs through bandpass filtering, noise removal, and baseline drift correction to support downstream analysis, and also extracts quantitative parameters such as mean amplitude and standard deviation:

$$(\text{ECG}_{\text{text}}, \text{ECG}_{\text{signal}}) = \text{ECGProcessor(ECGs)}.$$

*Electrophysiologists.* This functionality is implemented with NeuroKit2, a Python toolbox, to obtain 12-leads ECG measurements, include signal quality scores, heart rate variability (HRV) features, wave durations (e.g., QRS, PR, QT intervals), and extract heartbeat images from representative leads (e.g., II, I, and V5):

$$(\hat{E}, \hat{B}) = \text{NeuroKit2(ECGs)},$$

where $\hat{E} = \{\hat{e}_1, \ldots, \hat{e}_m\}$ represents the extracted ECG measurements, and $\hat{B} = \{\hat{b}_I, \hat{b}_{II}, \hat{b}_{V5}\}$ denotes the extracted heartbeat images from the respective leads.

*Echocardiography technician.* This tool functions as a view classifier (Vukadinovic et al., 2024) to extract standard cardiac views, including apical two-chamber (A2C), apical three-chamber (A3C), apical four-chamber (A4C), apical five-chamber (A5C), apical Doppler (AD), colour Doppler parasternal long-axis (DPL), colour Doppler parasternal short-Axis (DPS), parasternal long-axis (PSL), parasternal short-axis (PSS), suprasternal short-axis (SSN), and subcostal view (Sub), from raw DICOM data:

$$View = \text{ViewClassifier(DICOM)}.$$

*Echocardiography segmenter.* This tool performs segmentation for echocardiograms (ECHOs), which is essential for tracking cardiac function in clinical practice. Here, a segmentation network (Zhang et al., 2024) is employed to generate pixel-wise masks ($Mask$) to delineate cardiac structures in apical four-chamber videos (EchoVideo$_{A4c}$):

$$Mask = \text{SegNetwork(EchoVideo}_{A4c}).$$

*Cardiology fellow.* A fine-tuned multimodal model is employed for preliminary disease diagnosis ($Y$) based on diverse data modalities:

$$Y = \text{TGMM}(\text{Labs}, \text{ECGs}, \text{ECHOs}).$$

## Appendix C. Dataset

### C.1. MIMIC-IV Data

In this study, the ICU module is excluded, and focus is placed on hospital stay records from 223,452 patients to ensure data stability and relevance (Johnson et al., 2023). After applying stringent exclusion criteria, such as removing hospital stays without diagnostic outcomes, a refined subset of 1,524 samples is used. This subset includes 12-lead ECGs, echocardiograms, and laboratory test results, comprising 708 patients with prevalent heart failure (HF) and 816 without prevalent HF, as determined by ICD-9/10 diagnostic codes (Hong and Zeng, 2023). The dataset is then split using iterative stratification at a ratio of 5:1:1 (Sechidis et al., 2011). The fine-tuned models are trained and validated on the training and validation sets, while all other methods are evaluated on test set comprising 305 samples.

This dataset comprises laboratory test results, ECGs, and ECHOs. Laboratory measurements include key biomarkers such as Anion Gap, Bicarbonate, Creatinine, Potassium, and Sodium, supplemented by patient metadata such as age, ethnicity, gender, medical history, medication history, BMI, height, and weight. Missing values are left as missing (i.e., no imputation is performed). All ECGs are 12-lead, 10 seconds in duration, and sampled at 500 Hz, while echocardiogram data are stored in their raw Digital Imaging and Communications in Medicine (DICOM) format.

### C.2. PTB-XL and PTB-XL+ Data

The PTB-XL dataset contains 21,837 12-lead ECG recordings from 18,885 patients, each lasting 10 seconds and sampled at 500 Hz (Wagner et al., 2020). Complementarily, the PTB-XL+ dataset provides extracted ECG features along with key patient metadata such as gender and age (Strodthoff et al., 2023). These datasets are merged using patient identifiers and are used for myocardial infarction (MI) diagnosis. Each recording is independently annotated by two cardiologists, who assign probabilistic diagnostic labels, resulting in 9,514 Normal and 5,469 MI cases. For this study, only recordings with a 100% diagnostic probability for MI are included, yielding a final dataset of 7,172 Normal and 2,975 MI samples, totaling 10,147 recordings. Following official dataset guidelines, the data are split into training (8,167 samples), validation (991 samples), and test (989 samples) sets. The fine-tuned model is trained and validated on the training and validation sets, while all comparative methods are evaluated on the independent test set.

### C.3. PTB Diagnostic ECG Data

This dataset contains 549 ECG records collected from 290 subjects, aged between 17 and 87 years (mean age 57.2) (Goldberger et al., 2000). Each record comprises 15 simultaneously measured signals, including the standard 12-lead ECG (I, II, III, aVR, aVL, aVF, V1–V6)

and 3 Frank leads (Vx, Vy, Vz), all sampled at 1000 Hz with 16-bit resolution over a $\pm 16.384$ mV range. Diagnostic classes are available for 268 subjects. Since no fine-tuning is performed on this dataset, it is directly employed to evaluate the generalization capability of the fine-tuned model, with all samples used across all methods. Note that only the 12-lead ECG signals are utilized, not 3 Frank leads.

## Appendix D. Implementation Details

To uphold stringent data security and ensure full compliance with clinical governance frameworks, all models are deployed and executed entirely within on-premise infrastructure, with no reliance on external APIs. This ensures that patient data remains strictly within institutional boundaries, enabling secure inference in a fully controlled and auditable environment.

### D.1. CardAIc-Agents

All experiments involving CardAIc-Agents, such as main experiments, ablation studies, and parameter analyses, are conducted on a system equipped with three NVIDIA A100-SXM4 GPUs, each with 80GB of memory.

Both the guideline generation for the CardiacRAG Agent and the Chief Cardiologist rely on the DeepSeek-R1-Distill-Qwen-32B model, a distilled variant of the DeepSeek-R1 architecture designed to deliver efficient yet robust language generation. Following official guidelines, the model is configured with a temperature of 0.6 to balance creative variability with output consistency, and a maximum generation length is limited to 1024 tokens to ensure sufficiently detailed responses. Additional settings include: top-k is set to 40; top-p and typical-p are both set to 1.0; and a repetition penalty of 1.1 is applied to reduce output redundancy and enhance generation diversity.

For the multidisciplinary discussion team, two expert models are implemented. The first, MedGemma, which is based on Google's MedGemma-27b-it model (Sellergren et al., 2025), specialized in medical image analysis. The second expert, Qwen2.5-VL from Alibaba (Bai et al., 2023), is optimized for video processing tasks. Both models are configured with reasoning capabilities enabled and employ similar generation settings: a very low temperature of 0.01 to ensure highly focused and deterministic outputs, a maximum token generation limit of 256, and sampling parameters including a top-k of 40, top-p and typical-p set at 1.0, along with repetition penalties of 1.1 to reduce redundancy. Device allocation is set to automatic to facilitate efficient resource management. Note that, since MedGemma could only process images, frames from each echocardiogram view are concatenated into a single image to capture both spatial and temporal information across frames.

### D.2. Prevent Inference Failures

Test-time scaling (TTS) encompasses strategies that allocate additional computational budget during inference to improve the reliability and accuracy of model predictions. Recent work shows that increasing test-time compute, such as sampling multiple reasoning trajectories, deepening intermediate deliberation, or ensembling candidate outputs, systematically improves performance on reasoning intensive tasks (Muennighoff et al., 2025; Zhang et al., 2025a). At its core, TTS allows inference time computation to be adaptively increased to

correct model failures or explore alternative reasoning pathways without modifying model parameters.

Within our workflow, a practical source of instability stems from intermittent generation of ill-formed intermediate outputs, including missing mandatory fields, malformed JSON structures, or incomplete action specifications. Such structural defects interrupt downstream execution and may propagate through the multi-agent pipeline, ultimately compromising the validity of the final decision. To mitigate these failure modes, we introduce a lightweight adaptive TTS mechanism in which the agent automatically regenerates its output whenever structural or parsing constraints are violated (e.g., missing keys or JSON decoding errors). This procedure dynamically allocates additional inference compute only when needed, thereby instantiating the central principle of TTS: compute is used adaptively to ensure output correctness and procedural stability.

### D.3. Prompt

*Prompt Details.* Below, we present the prompt details for constructing our CardAIc-Agents workflow.

---

*PlanPrompt (System)*
You are a cardiac task planner. Analyze the given cardiac related task and produce a clear stepwise plan. Each step must include a brief "step" description and "tools_names" chosen only from the provided resource list. The plan must be returned as a single valid JSON object describing all steps, without repeated steps or tool calls, and without nested JSON, arrays, or markdown.

*PlanPrompt (User)*
Task: {["task_name"]}. Available data modalities: {dataset_map[["dataset"]]}. Available Resources/Tools: {tools_info}. Please provide a complete stepwise plan for this task and indicate which tool to use at each step, using only tools in {tools_all}.

---

*UpdatePrompt (System)*
You are the most authoritative cardiology expert. After each step, review all previous steps and all tool outputs to decide whether the current evidence is sufficient to give a final diagnosis for the task. If it is sufficient, give a concise conclusion and probability; if not, summarize what has been obtained so far and explain what additional information is needed. Always return one JSON object with fields: "conclusion", "answer" (a value between 0 and 1 for the probability of {task_name}), "action" ("stop" or "continue"), and "next_step" (null if you will follow the original next planned step {next_planned_step}, or a JSON object describing a new step using tools from {tools_all}). Use "stop" only when the diagnosis is clearly definitive.

*UpdatePrompt (User)*
Please review the following information and decide whether the current evidence is sufficient to provide a final decision for {["task_name"]}: {logs}. Return a single JSON object with keys "conclusion", "answer", "action", and "next_step". If you decide to continue with the original planned next step {next_planned_step}, set "next_step" to null; if you propose a different step, provide it as a JSON object. Do not assume or invent any parameters or test

results that are not explicitly mentioned.

---

*Complexity Level Prompt (System)*
You are a cardiology expert whose role is to determine the ComplexityLevel of a cardiac related task. Treat the case as a typical presentation unless otherwise stated. A task is "basic" if it has a clear diagnostic pipeline, and "advanced" if it is case dependent or requires expert judgment beyond standard diagnostic criteria. You must respond with a JSON object containing a single field "complexity" set to "basic" or "advanced".

*Complexity Level Prompt (User)*
Given the following cardiac related medical task and the complexity guidelines, determine whether the task is basic or advanced under a typical presentation assumption: {task_description}. Respond with a JSON object that contains only the field "complexity" with value "basic" or "advanced".

---

*Note.* The prompts listed here summarize the logic of each component. The complete prompt templates, with exact formatting and instructions, are provided in the code implementation.

## D.4. Baseline Models

*VLMs and variants.* All VLMs and their variants are configured with consistent generation settings to ensure a fair comparison. A low temperature of 0.01 is applied to encourage deterministic outputs, top-k fixed at 40, a repetition penalty of 1.1, and a maximum generation length of 4086 tokens. Sampling is enabled in all cases. LLaVA-Med is based on llava-med-v1.5-mistral-7b (set to 1024 tokens), while MedGemma is built upon medgemma-27b-it. The variants, referred to as MedGemma (CoT and ReAct), employ the same configurations but incorporate Chain-of-Thought (Wei et al., 2022) and ReAct (Yao et al., 2023) prompting strategies, respectively, to facilitate multi-step clinical reasoning and tool usage. Note that the ReAct prompting is implemented via LangChain, utilizing the same set of tools and tool descriptions as those adopted in the proposed CardAIc-Agents.

*Medical agents.* ReConcile (Chen et al., 2024) and MDAgents (Kim et al., 2024) are implemented using the same base models as CardAIc-Agents, including DeepSeek-R1-Distill-Qwen-32B, MedGemma-27b-it, and Qwen2.5-VL, and are configured with generation parameters identical to those of CardAIc-Agents to ensure consistency. In contrast, MedAgents (Tang et al., 2024) relies on a single large model performing three distinct roles, for which MedGemma-27b-it is employed. To guarantee a fair comparison, ECG data are preprocessed using the tool of Electrophysiologists from CardAIc-Agents to convert the signals into textual representations, which are subsequently fed into these agents. All experiments are conducted on the same hardware setup, using two or three NVIDIA A100-SXM4 GPUs, each with 80GB of memory.

*Fine-tuned VLMs.* Fine-tuning of larger models, such as Qwen2.5-VL-3B-Instruct (Bai et al., 2023) and Janus-Pro -7B (Chen et al., 2025), is performed using four NVIDIA A100-SXM4 GPUs, each equipped with 40GB of memory. Training leverages the Adafactor optimizer in conjunction with a ReduceLROnPlateau learning rate scheduler to adaptively adjust the learning rate based on validation performance. Mixed precision training is utilized

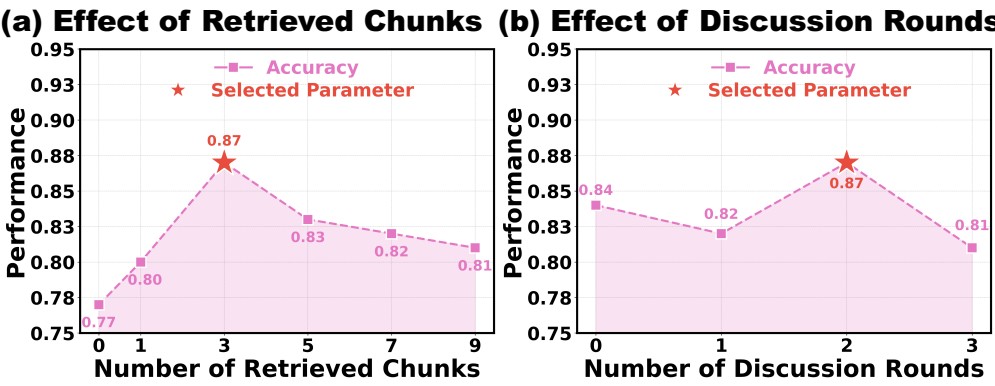

Figure S1: Hyperparameter selection on the MIMIC-IV dataset.

to enhance computational efficiency. Low-Rank Adaptation (LoRA) is applied to reduce trainable parameters and improve training efficiency without sacrificing model performance.

## Appendix E. Additional Experiments and Analysis

### E.1. Parameter Analysis

The analysis was further extended to assess the sensitivity of two key parameters: the number of retrieved chunks and the maximum number of rounds allowed for the multi-disciplinary discussion (see Figure S1). MIMIC-IV was used as a typical example for this analysis. The results indicated that increasing the number of retrieved chunks initially improved performance, peaking at three chunks. Beyond this point, performance declined. This trend could be attributed to two factors: when too few knowledge chunks were available, the model lacked sufficient context to make informed decisions; however, when the number exceeded a certain threshold, the input might have surpassed the effective token processing capacity of the model, which degraded performance. Similarly, the analysis revealed that the best accuracy was achieved with two rounds of discussion. Both fewer and more rounds resulted in performance below the baseline (i.e., without the discussion tool). This highlighted the importance of carefully tuning the number of discussion rounds, as suboptimal settings could lead to unreliable intermediate reasoning and potentially misguide the final decision made by the chief cardiologist.

### E.2. Confusion Matrices

The confusion matrix analysis in MIMIC-IV showed that the CardAIc-Agents delivered more balanced and reliable classification outcomes compared with the best-performing VLM (MedGemma) and the best-performing agent (MedAgents) baseline (Figure S2). Specifically, the CardAIc-Agents achieved 161 true negatives and 103 true positives, whereas the MedGemma exhibited substantially higher error counts, including 27 false positives and 47 false negatives. The MedAgents similarly demonstrated poorer performance, producing 47 false positives and 32 false negatives. As shown in the figure, the marked reduction in both false positives and false negatives highlighted the superior discriminative capability of

the CardAIc-Agents, particularly in mitigating missed positive cases, thereby improving its potential clinical utility.

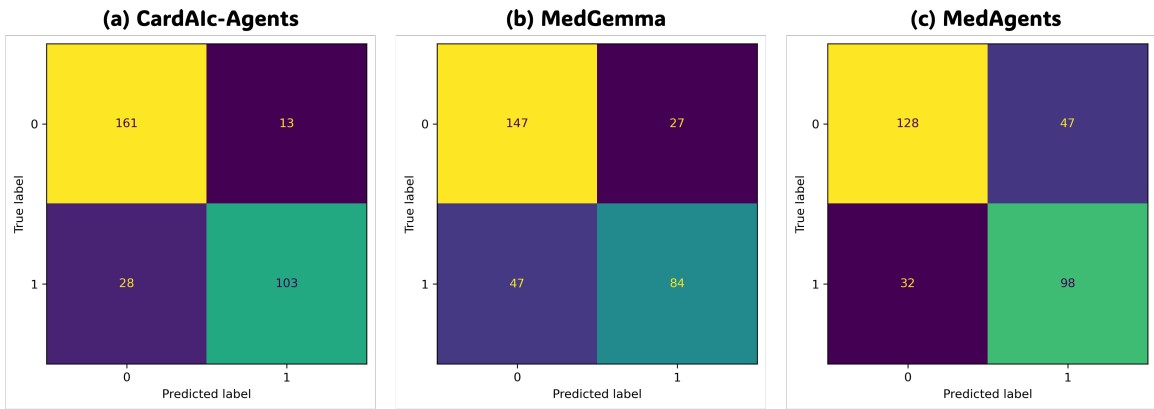

Figure S2: Confusion Matrices: (a) CardAIc-Agents; (b) MedGemma (best-performing VLM baseline); (c) MedAgents (best-performing agent baseline).

### E.3. Case Study

Figure S3 presented a case study that illustrated the operational workflow of CardAIc-Agents, emphasizing its adaptive capability to dynamically adjust its approach based on the input data. Depending on the analysis outcomes, the system could either continue with the current plan, modify its strategy, or halt further processing. This case study effectively showed the flexibility of CardAIc-Agents in managing complex and evolving cardiac tasks. Note that this case was processed in 56.93 seconds using three NVIDIA A100-SXM4 GPUs.

### E.4. Retrieved Knowledge Quality

The proposed hybrid retrieval method was further evaluated by comparing its retrieval quality against TF-IDF and DPR baselines. Retrieval performance was assessed using accuracy and the average redundancy score (AVG), where lower redundancy indicates less duplicated or irrelevant retrieved content. As shown in Table S1, TF-IDF achieved an accuracy of 0.83 with an average redundancy of 0.91, while DPR achieved 0.83 with an average redundancy of 0.96, indicating substantial redundancy in retrieved passages. In contrast, our method obtained the highest accuracy of 0.87 and produced the lowest redundancy (0.83), demonstrating that the hybrid retrieval design retrieved more relevant and diverse evidence.

### E.5. Sensitive Analysis of Task Complexity Assessment

To examine the robustness of CardAIc-Agents in complexity assessment, a prompt sensitivity study was conducted. Specifically, four additional participants were invited to design alternative complexity assessment prompts for HF and AF detection (details below). Across

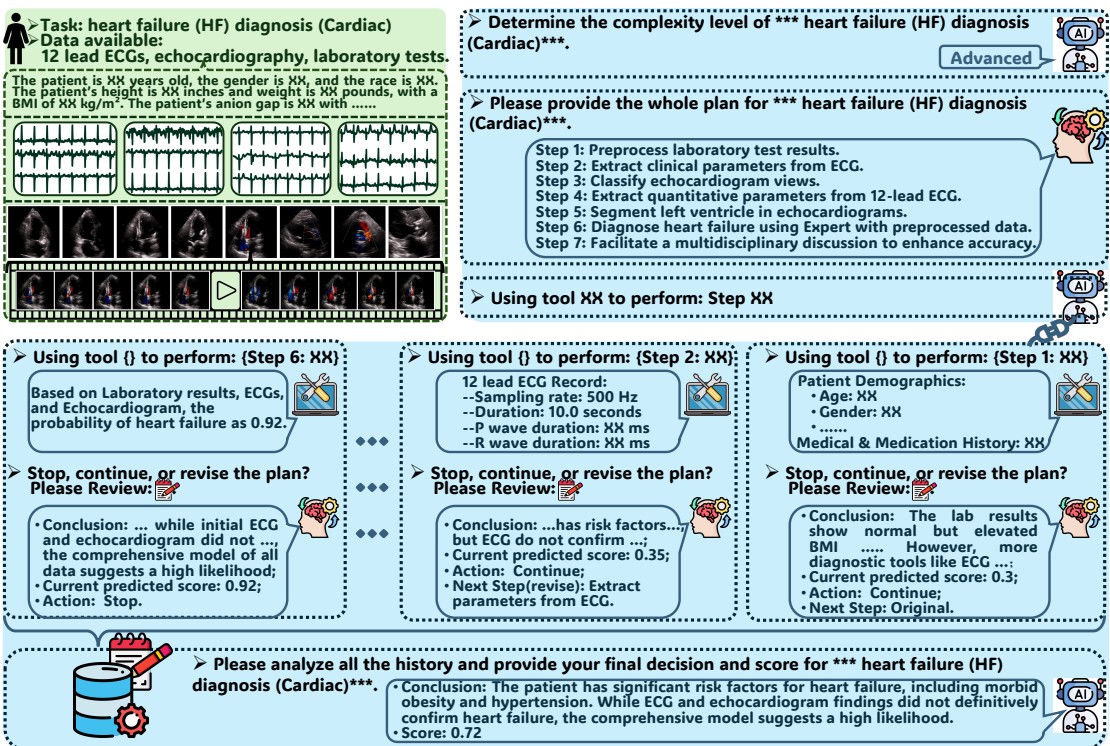

Figure S3: Illustration of the case study that shows the end-to-end operational workflow of CardAIc-Agents, spanning user query intake, internal multimodal reasoning, and final decision generation. The internal reasoning trace exemplifies the stepwise update mechanism, in which the agent may (i) continue executing the current plan, (ii) revise the plan, or (iii) terminate execution prior to completing the originally proposed steps. For clarity, only representative reasoning steps are shown; The symbol ▶ denotes prompt (system or user); xx denotes the content of each reasoning step, and {} specifies the tool invoked at that step.

Table S1: Retrieval quality comparison using accuracy and average redundancy (AVG).

| Method | ACC | AVG |
|---|---|---|
| TF-IDF | 0.83 (0.79, 0.87) | 0.91 (0.91, 0.92) |
| DPR | 0.83 (0.78, 0.87) | 0.96 (0.96, 0.96) |
| Ours | 0.87 (0.83, 0.90) | 0.83 (0.81, 0.84) |

all prompts (including the original prompt), the complexity labels were consistent, classifying HF as advanced and AF as basic, aligning with commonly reported clinical viewpoints (Heidenreich et al., 2022; , UK; van Vliet et al., 2025).

Here, predefined complexity criteria are included in the prompt to reduce sensitivity: 1) basic: the task has a clear diagnostic pipeline. 2) advanced: the task is complex and case-dependent or requires expert judgment beyond standard diagnostic criteria. In addition, the impact of misclassification was further evaluated under a worst case setting, where an advanced case was incorrectly routed to the basic workflow. Results on the MIMIC-IV dataset showed that accuracy was 0.86 when the case was routed to the basic workflow, and 0.89 when routed to the advanced workflow, remained higher than all baseline methods.

*Case 1: Complexity Level Prompt (User)*
Given the following cardiac related medical task and the complexity guidelines, determine whether the task is basic or advanced under a typical presentation assumption: {task description}. Complexity Guidelines: 1) basic: the task has a clear diagnostic pipeline. 2) advanced: the task is complex and case-dependent or requires expert judgment beyond standard diagnostic criteria. Respond with a JSON object that contains only the field "complexity" with value "basic" or "advanced".

*Case 2: Complexity Level Prompt (User)*
You will be given a cardiac related medical task plus the complexity guidelines. Under a typical presentation assumption, decide whether this task should be labeled as basic or advanced. Task: {task description}. Complexity Guidelines: 1) basic: the task has a clear diagnostic pipeline. 2) advanced: the task is complex and case-dependent or requires expert judgment beyond standard diagnostic criteria. Output: Return only one JSON object with exactly one field: {"complexity":"basic"} or {"complexity":"advanced"}.

*Case 3: Complexity Level Prompt (User)*
Based on the complexity guidelines, classify the following cardiac related medical task as either basic or advanced, assuming a typical clinical presentation: {task description}. Complexity Guidelines: 1) basic: the task has a clear diagnostic pipeline. 2) advanced: the task is complex and case-dependent or requires expert judgment beyond standard diagnostic criteria. Respond with a single JSON object containing only the key "complexity" and the value "basic" or "advanced".

*Case 4: Complexity Level Prompt (User)*
Determine the complexity category for the cardiac related task below using the provided guidelines. Assume a typical patient presentation. Task description: {task description}. Complexity Guidelines: 1) basic: the task has a clear diagnostic pipeline. 2) advanced: the task is complex and case-dependent or requires expert judgment beyond standard diagnostic

criteria. Return ONLY valid JSON with one field named "complexity" and a value of "basic" or "advanced".

