# OpenReview forum: "CardAIc-Agents: A Multimodal Framework with Hierarchical Adaptation for Cardiac Care Support"
_MIDL.io/2026/Conference — MIDL 2026 Poster_

### Official Review · Reviewer_ggGy · 2026-01-04

**Confidence:** 3
**Preliminary Rating:** 4
**Final Rating:** 4

**Summary:**

This paper presents CardAIc-Agents, a multimodal multi-agent framework designed to automate cardiac care tasks through hierarchical adaptation. A dedicated CardiacRAG agent utilizes a hybrid retrieval mechanism to generate task-specific plans from a curated cardiac knowledge base, while a Chief agent orchestrates external tools for execution and dynamic refinement. Complex cases trigger a simulated multidisciplinary discussion team and visual review panels are generated to assist clinician verification. Evaluation across three datasets (MIMIC-IV, PTB-XL, PTB Diagnostic) demonstrates superior performance compared to existing medical VLMs and agentic baselines.

**Strengths:**

- Integrating specific cardiac tools like echocardiogram segmentation and ECG signal processing allows the system to ground its reasoning in actual clinical data rather than relying solely on latent knowledge.
- Hybrid retrieval mechanism within the CardiacRAG module effectively addresses the limitations of standard semantic search by incorporating domain-specific keyword filtering.
- Proposed adaptive workflow that distinguishes between basic and advanced complexity adds a layer of efficiency and mimics clinical triage processes.
- Inclusion of a visual review panel for clinician verification is a thoughtful design choice that significantly enhances the potential for real-world trust and deployment.
- Extensive experiments across multimodal datasets (MIMIC-IV, PTB-XL) provide convincing evidence of the framework's superiority over general-purpose medical VLMs.

**Weaknesses:**

- Computational cost and latency are significant concerns, as the iterative "discussion" and multiple model calls described in the case study took nearly a minute on high-end hardware, which may hinder real-time deployment.
- Mechanism for complexity assessment relies on a simple prompt to the LLM, which may be fragile and lacks a robust quantitative basis for determining when to switch workflows.
- Comparison with baseline VLMs feels slightly unbalanced if those models were not provided with the same pre-processed tool outputs (e.g., extracted ECG parameters) that the agent system enjoyed.
- Dependence on the quality of the underlying "authoritative" knowledge base is critical, yet the process for automatic curation and conflict resolution between sources is under-explained.
- False negative rate, while improved over baselines, remains a concern for a system designed to support critical cardiac diagnoses, raising safety questions.

**Detailed Comments:**

- Figure 1 is quite dense; breaking the workflow into two distinct diagrams for the RAG process and the Agent execution might improve readability.
- Clarify how the "consistency" in the multidisciplinary discussion is measured—is it just a text output, or is there a structured voting mechanism?
- Please expand on how the system handles scenarios where external tools (e.g., the segmentation network) produce low-confidence or erroneous outputs.
- Section 3.3 would benefit from a more granular analysis of which specific tools contributed most to the performance gains in the "Advanced" workflow.
- Formatting of the references needs standardization, as there are inconsistencies in capitalization and journal names.

**Justification Of Final Rating:**

Addressed most my concerns (prompt study validates complexity assessment; MDT breakdown justifies overhead; baselines with identical inputs confirm adaptive reasoning drives gains; ablations disentangle contributions. Hierarchical adaptation clarified. Despite latency, clinical value warrants acceptance), and I beleive this paper reached the bar of acceptance.

**Justification Of The Preliminary Rating:**

- This paper represents a solid engineering effort that successfully integrates recent advances in agentic workflows with domain-specific medical tools.
- Proposed hierarchical approach addresses the "one-size-fits-all" limitation of previous medical agents and the visual outputs are highly relevant for the MIDL community.
- While there are concerns about inference latency and the simplicity of the complexity classifier, the overall contribution to multimodal cardiac care support is valuable and well-validated.

**Questions To Address In The Rebuttal:**

- How sensitive is the complexity assessment module to variations in prompting, and did you evaluate the accuracy of this classification against human judgment?
- Can you provide a more detailed breakdown of inference time versus improvement in accuracy to justify the computational overhead of the multidisciplinary discussion?
- Were the baseline VLMs provided with the exact same textual representations of the ECG/Echo data as the agents, or did they have to process raw data embeddings?

---

> ### Author Response · Authors · 2026-01-24
>
> **Q1. Sensitivity of Complexity Assessment Module and Validation**
>
> Thank you for the comment. We have further performed a prompt sensitivity study to examine its robustness. Specifically, we invited four participants to design alternative complexity assessment prompts for HF and AF detection. Across all prompts (including the original prompt), CardAIc-Agents produced consistent labels: HF as advanced and AF as basic, consistent with clinical viewpoints [1,2] (prompts in Appendix E5).
>
> In this work, we had reduced sensitivity by explicitly predefining the complexity criteria in the original prompt: 1) basic: the task has a clear diagnostic pipeline; 2) advanced: the task is complex and case dependent or requires expert judgment beyond standard diagnostic criteria. Meantime, we had quantified the impact of misclassification: an advanced case is incorrectly routed to the basic workflow. Even then, performance had dropped modestly (AUC 0.89 to 0.87; Tab. 2) and had remained higher than all baselines.
>
> In addition, although task complexity was not formally validated, the output aligned with clinical viewpoints in the literature: HF assessment may require additional tests on a case by case basis [1], whereas AF detection follows a relatively standardized paradigm [2].
>
> [1]Journal of the American College of Cardiology,2022,79(17):263-421
>
> [2]npj Digital Medicine,2025,8(1):177.
>
> **Q2. Time-accuracy (MDT)**
>
> Thank you for the suggestion. We now report a time–accuracy breakdown for the MDT in the table below (Tab.3 in the manuscript). On MIMIC-IV, MDT was triggered for only 6\% of samples; even with this low trigger rate, enabling MDT increases overall accuracy from 0.84 to 0.87, with mean latency rising from ~50s to ~79s.
>
> On the triggered subset only, MDT yielded larger gains (0.84 vs. 0.74), at the cost of higher per-case latency (~194s vs ~52s). This highlights that on-demand MDT yields performance improvements at the cost of latency overhead on a small subset of samples. However, this trade-off parallels clinical escalation workflows, in which multidisciplinary discussion is reserved for a minority of complex cases and incurs substantial time overhead.
>
> |Method|Mean(s)|P95(s)|ACC|AUC|Trig_Rate(%)|Trig_Mean(s)|Trig_P95(s)|Trig_ACC|Trig_AUC|
> |:-:|:-:|:-:|:-:|:-:|:-:|:-:|:-:|:-:|:-:|
> |No MDT|50.44|62.98|0.84|0.88|0|52.72|59.81|0.74|0.55|
> |On-demand MDT|79.14|161.01|0.87|0.89|6|194.31|285.25|0.84|0.61|
>
> **Q3. ECG/Echo Preprocessing**
>
> For a fair comparison, all baselines should use the same inputs, task outputs, and evaluation protocol as our method, with variation only in the underlying models. However, current VLMs can't directly consume raw ECG (.dat) or Echo (DICOM) inputs, whereas ours can invoke dedicated tools to process them. To ensure VLM compatibility, we therefore used the same preprocessing tools as in our agent to convert inputs into VLM-compatible modalities: ECG and blood tests (Lab) were converted into text; Echo was converted into images. Thus, VLM baselines received text extracted from ECG and Lab, images from Echo, using the same preprocessing tools as our method.
>
> Nevertheless, we feed the full tool outputs manually into the baselines to assess their ability to directly leverage the provided evidence (see Table below). All baselines exhibit consistent improvements compared with the setting that uses only preprocessing tools. Despite these improvements under manual provision of tool outputs, none of the baselines surpasses ours with automatic tool invocation. This result confirms that simply exposing baselines to all tool outputs is insufficient to close the performance gap.
>
> Overall, we view the domain-specific tools as integral to CardAIc-Agents, which can automatically leverage them to produce reliable predictions. In contrast, the baselines can't operate on raw inputs, and a clear performance gap remains even when preprocessing tools or all tool outputs are provided to them.
>
> |Method|Limited Tools_ACC|Limited Tools_AUC|All Tools_ACC|All Tools_AUC|
> |:-:|:-:|:-:|:-:|:-:|
> |LLaVA-Med|0.35|0.34|0.64|0.77|
> |MedGemma|0.76|0.82|0.76|0.83|
> |MedGemma(CoT)| 0.65 |0.81|0.77|0.84|
> |MedGemma(ReAct)|-|-|0.67|0.71|
> |Ours|-|-|0.87|0.89|
> *Limited Tools=preprocessing tools only*
>
> **Low-confidence or erroneous outputs from tools:**
>
> The adaptive test-time scaling mechanism was designed for procedural robustness (Appendix D.2), and final decision relied on multiple evidence sources rather than any single tool output to limit the influence of isolated tool errors.
>
> **We have added a tool ablation study as Tab.2 in the revised manuscript:**
> |Electrophysiologists|Echo Segmenter|Cardiology Fellow|MDT|ACC|AUC|
> |:-:|:-:|:-:|:-:|:-:|:-:|
> ||✓|✓|✓|0.84|0.88|
> |✓|| ✓ |✓|0.83|0.88|
> |✓|✓||✓|0.77|0.84|
> |✓|✓|✓||0.84|0.88|
> |✓|✓|✓|✓|0.87|0.89|
>
> The revised version also clarifies MDT consistency, revises Fig.1, and corrects the references, with changes highlighted in blue. We thank the reviewer for their valuable feedback.

---

> > ### Comment · Reviewer_ggGy · 2026-01-28
> >
> > Thanks for your detailed response, I will keep my postive ratings.

---

### Official Review · Reviewer_Gv5N · 2026-01-07

**Confidence:** 3
**Preliminary Rating:** 4

**Summary:**

This paper presents CardAIc-Agents, a multimodal agentic framework designed to support cardiac diagnostic workflows by integrating domain-specific retrieval, adaptive task planning, tool orchestration, and multi-agent discussion. The experiments show that CardAIc-Agents consistently outperforms baselines across datasets.

**Strengths:**

-	The workflow closely mirrors real cardiac diagnostic processes (iterative testing, evidence accumulation, MDT-style reasoning).
-	The separation between knowledge-driven planning (CardiacRAG) and execution/decision-making (Chief Agent) is conceptually clean and well justified.
-	The system meaningfully integrates signal processing, segmentation, and view classification rather than relying solely on language reasoning.

**Weaknesses:**

-	I am concerned that some improvements may stem from the inclusion of strong domain-specific tools rather than the adaptive agentic reasoning itself, which is not fully disentangled.
-	Complexity assessment (basic vs. advanced) is heuristic and not quantitatively validated. How sensitive is performance to errors in the initial complexity assessment?
-	Runtime cost and deployment constraints can be discussed.

**Detailed Comments:**

See the above questions. The overall presentation is clear.

**Justification Of The Preliminary Rating:**

This paper provides a well-structured, clinically realistic agentic framework that advances multimodal cardiac AI in a practical and interpretable direction. I also appreciate the current form but the paper can be improved by clearly answering the above questions.

**Questions To Address In The Rebuttal:**

-	I am concerned that some improvements may stem from the inclusion of strong domain-specific tools rather than the adaptive agentic reasoning itself, which is not fully disentangled.
-	Complexity assessment (basic vs. advanced) is heuristic and not quantitatively validated. How sensitive is performance to errors in the initial complexity assessment?
-	Runtime cost and deployment constraints can be discussed.

---

> ### Author Response · Authors · 2026-01-24
>
> **Q1. Improvements from Tools and Adaptive Agentic Reasoning**
>
> Thank you for the comment. We agree that domain specific tools contribute to performance. However, our results indicate that the adaptive agentic reasoning also provides additional gains beyond the tools alone. Specifically, in Tab.2, under the same set of domain specific tools, removing the adaptive reasoning component reduces accuracy from 0.87 to 0.80 (p < 0.05).
>
> We also provide new experimental results in the table below (and have updated Tab.2 in the manuscript) to disentangle the contribution of each domain-specific tool. Ablating any single tool consistently reduces performance, with an accuracy drop of at least 0.03 when the Electrophysiologist tool is removed (p < 0.05).
>
> |Electrophysiologists|Echo Segmenter|Cardiology Fellow|MDT|ACC|AUC|
> |:-:|:-:|:-:|:-:|:-:|:-:|
> ||✓|✓|✓|0.84|0.88|
> |✓|| ✓ |✓|0.83|0.88|
> |✓|✓||✓|0.77|0.84|
> |✓|✓|✓||0.84|0.88|
> |✓|✓|✓|✓|0.87|0.89|
>
> Therefore, both tools and adaptive agentic reasoning itself have made contributions.
>
> **Q2. Sensitivity of Complexity Assessment Module**
>
> Thank you for the comment. We agree that the initial *Basic* vs. *Advanced* complexity assessment is heuristic. In our implementation, task complexity was discussed with clinicians. Plus, the result from our CardAIc-Agents was also consistent with reported clinical viewpoints in the literature, where HF assessment may require additional tests on a case-by-case basis [1], whereas AF detection follows a relatively standardized paradigm [2].
>
> To examine the robustness of CardAIc-Agents in complexity assessment, we had further performed a prompt sensitivity study. Specifically, we invited four additional participants to design alternative complexity assessment prompts for HF and AF detection. Across all prompts (including the original prompt), CardAIc-Agents produced consistent complexity labels, classifying HF as advanced and AF as basic, in line with commonly reported clinical viewpoints (prompts are provided in Appendix~E5).
>
> In this work, we had reduced sensitivity by explicitly predefining the complexity criteria in the original prompt: 1) basic: the task has a clear diagnostic pipeline; 2) advanced: the task is complex and case dependent or requires expert judgment beyond standard diagnostic criteria. Meantime, we had quantified the impact of misclassification: an advanced case is incorrectly routed to the basic workflow. Even then, performance had dropped modestly (AUC 0.89 to 0.87; Tab. 2) and had remained higher than all baselines.
>
> [1]Journal of the American College of Cardiology,2022,79(17):263-421
>
> [2]npj Digital Medicine,2025,8(1):177.
>
> **Q3. Runtime Cost and Deployment Constraints**
>
> We now include an explicit cost comparison against VLMs and their variants, agentic systems, and fine-tuned baselines in the table below (Tab.1 has been updated in the manuscript). The results show that among VLMs, simple inference achieves faster runtime but substantially lower accuracy (e.g., 0.35 for LLaVA), whereas variants such as CoT and ReAct exhibit substantially higher latency and lower accuracy. For agentic systems, MDAgents runs faster (\~64s vs. \~79s) but achieves lower performance (0.52 vs. 0.87), whereas other agents are both slower (\~157s for MedAgents and ~103s for ReConcile) and less accurate than our method. Finally, fine-tuned model only take around \~1s for inference only using an already trained fine-tuned model (\~43 hours for Janus-Pro fine-tuning on the PTB-XL dataset); however, fine-tuned VLMs are typically task-specific, with limited generalization, flexibility, and interpretability.
>
> Although CardAIc-Agents is designed to support realistic clinical use (e.g., processing raw inputs, adaptively invoking tests or MDT), these capabilities introduce additional computational steps and increase overall system requirements, making higher runtime a potential deployment constraint. However, the improved accuracy can be valuable in resource-limited settings, where clinician shortages make a slower but reliable diagnostic system practically useful. Future work will explore efficiency optimizations to mitigate latency overhead in clinical practice.
>
> |Category|Method|Mean (s)|P50 (s)|P90 (s)|P95 (s)|
> |:-:|:-:|:-:|:-:|:-:|:-:|
> |VLMs & variants|LLaVA Med|22.515|6.800|62.014|100.453|
> ||MedGemma|15.046|14.581|20.094|22.216|
> ||MedGemma (CoT)| 359.038|317.151|671.749|830.398|
> ||MedGemma (ReAct)|139.422|102.678|232.291|309.628|
> |Medical agents |MedAgents|156.870|126.788|256.343|291.306|
> ||ReConcile |103.206|93.582|154.986|167.699|
> ||MDAgents |64.563|77.454|88.368|90.715|
> |Fine-tuned VLMs|Qwen2.5 VL|1.173|1.163|1.183|1.185|
> ||Janus Pro|1.247|1.238|1.258|1.259|
> |Proposed|CardAIc-Agents|79.137|75.584|126.148|161.010|
>
> *Note: Latency is reported in seconds (s).*
>
> We thank the reviewer for their time and valuable feedback. All corresponding revisions are highlighted in blue in the revised manuscript.

---

> > ### Comment · Reviewer_Gv5N · 2026-02-02
> >
> > This paper provides a well-structured, clinically realistic agentic framework that advances multimodal cardiac AI in a practical and interpretable direction. I am satisfied with the rebuttal and maintain my acceptance attitude.

---

> > > ### Author Response · Authors · 2026-02-02
> > >
> > > Thank you for your helpful suggestions, which have improved and strengthened the manuscript. Thank you again for your time.

---

### Official Review · Reviewer_R1Dq · 2026-01-09

**Confidence:** 3
**Preliminary Rating:** 4
**Final Rating:** 5

**Summary:**

The authors introduce "CardAIc-Agents," a multimodal framework designed for cardiac care support. The system comprises two main agents: a CardiacRAG Agent that generates task plans using a specialized cardiac knowledge base (with hybrid retrieval), and a Chief Agent that orchestrates tool execution and adapts the workflow based on task complexity. The framework features a "Multidisciplinary Discussion Team" (MDT) for complex cases and generates visual panels for clinician verification. Experiments on three datasets (MIMIC-IV, PTB-XL, PTB Diagnostic) show the method outperforms general medical VLMs and existing agent frameworks.

**Strengths:**

- Comprehensive Clinical Workflow: The framework is well-aligned with real-world clinical needs. Features like the Visual Review Panel (generating evidence for verification) and the MDT (simulating board discussions) mimic hospital workflows better than standard "black box" classifiers.

- Strong Empirical Results: The method demonstrates superior performance compared to generalist VLMs and other agentic baselines. For instance, on MIMIC-IV, it achieves 87% accuracy compared to LLaVA-Med's 35%, highlighting the necessity of domain-specific tools over generic vision encoders.

- Adaptive Design: The strategy of assessing task complexity (Basic vs. Advanced) to trigger different planning depths (Static vs. Iterative) is a logical and efficient design choice.

**Weaknesses:**

- Buried Comparisons (Fine-tuning): The comparison against fine-tuned models (e.g., fine-tuned Janus-Pro) is relegated to Appendix E.4. As noted in Table S1, the fine-tuned Janus-Pro actually achieves higher AUC (0.99) than the proposed agent (0.96) on PTB-XL. This is an interesting finding: if a single fine-tuned model outperforms the complex multi-agent system, the justification for the agentic overhead becomes weaker. This comparison could require more discussion.

- Unclear Definitions ("Hierarchical"): The title and conclusion emphasize "Hierarchical Adaptation," but the main text never explicitly defines or discusses this hierarchy. Is it the hierarchy between Agents? Between complexity levels? This central concept is under-explained.

- Writing Style: The paper is written almost entirely in the simple past tense. In computer science and medical imaging venues like MIDL, the standard is the present tense for the proposed method. The current style makes it read like a clinical trial report rather than a method paper.

- Figure Clarity: Figure 1 is cluttered and difficult to parse, especially without explanations in the caption. Specific labels (like 'j') are referenced in the text but are missing in the diagram.

**Detailed Comments:**

- Writing Style: Please revise the paper to use the present tense when describing your methodology and architecture.

- Main Text vs. Appendix: Please revise which content of the appendix could be part of the main text (e.g. E.4).

- Clarification: Please define and discuss the "hierarchy" of your method.

- Figure 1: Please improve the legibility of the labels (e.g., 'j') and consider adding the MDT and review panel.
- Figure 3: The figure is difficult to comprehend without detailed explanations (e.g. in the caption).

**Justification Of Final Rating:**

The paper proposes a novel agentic framework for cardiac care. The authors successfully addressed my concerns regarding the comparison with fine-tuned models, added detailed cost analysis and clarified the idea behind the the hierarchical mechanisms. I hence raised my score to a strong accept.

**Justification Of The Preliminary Rating:**

The proposed framework is novel, comprehensive, and addresses the "black box" issue of standard medical AI. However, the manuscript hides comparisons (fine-tuning) in the appendix and suffers from stylistic issues (tense) and vague terminology ("hierarchical"). If the authors move the fine-tuning comparison to the main text and justify the latency trade-offs in the rebuttal, this could be a comprehensive and well written paper.

**Questions To Address In The Rebuttal:**

**Agent vs. Fine-Tuning:** Your Appendix shows that fine-tuned Janus-Pro achieves higher AUC than your agent on PTB-XL (0.99 vs 0.96). Given this, what is the specific advantage of your complex, multi-agent framework over a simpler fine-tuned model? (e.g., Is it interpretability? Flexibility?)

**Latency & Cost:** Appendix E.3 mentions a runtime of ~57 seconds for a single case. Is this practical for clinical deployment? Please provide a cost analysis (latency/computation) comparing your method to the faster fine-tuned baselines.

**Hierarchy Definition:** Can you clearly define what you mean by "Hierarchical Adaptation"? Is it the switch between "Basic/Advanced" modes, or something else?

---

> ### Author Response · Authors · 2026-01-24
>
> **Q1. Agent vs. Fine-Tuning**
>
> Thank you for the insightful question. The experiments did show that Janus-Pro fine-tuned on PTB-XL achieves a higher AUC for AF detection, which indicates that for a well-defined task on a fixed dataset, fine-tuned models may reach higher performance. However, fine-tuned models have several limitations.
>
> *Deployment Overhead.* Fine-tuned models typically require additional training overhead prior to deployment, whereas our agent is ready to use. Note that although LLMs can be adapted via prompting, this still constitutes a separate tuning step prior to deployment.
>
> *Generalization.* When evaluated out-of-distribution on a new cohort without further adaptation, fine-tuned models degrade more than our agent (e.g., 0.72–0.75 vs. 0.77; Tab.1), indicating better robustness to dataset shift.
>
> *Flexibility.* Fine-tuned models are bound to a fixed input schema and task objective after training, whereas our agent enables multi-task inference (e.g., diagnosis, view extraction, segmentation, detection of P, QRS, and T waves) with on-demand modality use within an adaptive workflow. That is, fine-tuning requires a separate procedure for each task, while our proposed agent can be used in various tasks without additional training or fine-tuning.
>
> *Interpretability.* Fine-tuned models largely rely on post-hoc explanations (e.g., SHAP, Grad-CAM), and correcting clinically misaligned reliance often requires additional fine-tuning or retraining. In contrast, our agent provides stepwise evidence and supports targeted updates via prompt refinement and, where applicable, knowledge-base updates.
>
> It is worth noting that the fine-tuned model’s higher AUC without a corresponding improvement in accuracy may reflect coarser probability outputs from LLM-based agents, which often yield elevated baseline scores without explicit diagnostic cues, whereas fine-tuned models produce more precise and better-calibrated estimates with higher numerical precision (e.g., 32-bit). Despite the lower AUC, the proposed agent showed notable accuracy gains, highlighting its potential for early screening, for example, in resource-limited settings.
>
> **Q2. Latency & Cost**
>
> Thank you for the comment. The reported \~57s runtime refers to the end-to-end execution time for the illustrative case in Fig.S3 and reflects the practical deployment setting of our current pipeline.
>
> We now include an explicit cost comparison against fine-tuned baselines in the table below and update Tab.1 in the manuscript. The results show that our agent has a mean end-to-end runtime of \~79s, compared with \~1.24s for inference only with an already trained fine-tuned model (\~43 hours for Janus-Pro fine-tuning on the PTB-XL dataset). The advantages of our agent over fine-tuned models are discussed in Q1.
>
> In addition, we compare runtime against other agent baselines: MDAgents runs faster (\~64s vs. \~79s) but achieves lower performance (0.52 vs. 0.87), while other agents are both slower and less accurate than our method. We also benchmark a range of VLMs and their variants; although some run faster, their accuracy is substantially lower, and MedGemma (CoT) incurs substantially higher latency (~359s).
>
>
> |Category|Method|Mean (s)|P50 (s)|P90 (s)|P95 (s)|
> |:-:|:-:|:-:|:-:|:-:|:-:|
> |VLMs & variants|LLaVA Med|22.515|6.800|62.014|100.453|
> ||MedGemma|15.046|14.581|20.094|22.216|
> ||MedGemma (CoT)| 359.038|317.151|671.749|830.398|
> ||MedGemma (ReAct)|139.422|102.678|232.291|309.628|
> |Medical agents |MedAgents|156.870|126.788|256.343|291.306|
> ||ReConcile |103.206|93.582|154.986|167.699|
> ||MDAgents |64.563|77.454|88.368|90.715|
> |Fine-tuned VLMs|Qwen2.5 VL|1.173|1.163|1.183|1.185|
> ||Janus Pro|1.247|1.238|1.258|1.259|
> |Proposed|CardAIc-Agents|79.137|75.584|126.148|161.010|
>
> *Note: Latency is reported in seconds (s).*
>
> **Q3. Hierarchy Definition**
>
> Switching between *Basic* and *Advanced* modes constitutes one component of our notion of Hierarchical Adaptation. In this work, Hierarchical Adaptation refers to a multi-level decision and refinement process that includes: (1) stratifying task complexity (e.g., *Basic* vs. *Advanced*),
> (2) iteratively refining plans as new evidence becomes available,
> (3) initiating team-level discussions  when the current evidence is insufficient to reach a final conclusion, and
> (4) generating visual outputs to support further validation and human oversight.
> Together, these hierarchical mechanisms allow the system to adapt its workflow to the requirements of specific tasks and individual patients. We have clarified this definition in the Methods section and thank the reviewer for highlighting the need for this distinction.
>
> We also revised the manuscript to address *Detailed Comments* on writing style, content organization, appendix cross-references in the main text, and figures, with all changes highlighted in blue. We thank the reviewer for their valuable feedback.

---

### Author Rebuttal · Authors · 2026-01-24

**Rebuttal:**

We appreciate the reviewers for their overall positive assessment, valuable suggestions, and detailed comments.

In the revised manuscript (attached and marked in blue), we have made the following updates:

* Clarified the MDT procedure and reported its time overhead (Table 3).
* Added latency comparisons for our method and baselines (Table 1).
* Moved the fine-tuned model comparison into the main text (Section 3.3).
* Added ablation studies of domain-specific tools (Table 2).
* Updated Figures 1 and Figures 3 and revised the captions for clarity.
* Clarified the definition of *hierarchical adaptation* and improved tense consistency, appendix cross-references, and reference formatting.

All individual reviewer comments are addressed below.

**Supporting Material:**

/attachment/3d9626046893326d3326be45db845634c9c211f3.pdf

---

### Comment · Area_Chair_sNCG · 2026-02-01
**Please enter Final Rating**

Dear reviewers,
Please note that today, Feb 1 is the last day to enter your final ratings. Thank you to those who have already updated. If you have not yet, please take a moment to look through the author’s rebuttal and update your final score and reasoning.
We greatly appreciate your important contribution to MIDL.
Thank you!
Your AC

---

### Meta-Review · Area_Chair_sNCG · 2026-02-09

**Recommendation:** Accept (Oral)
**Confidence:** 5

**Metareview:**

All reviewers agree that this paper meets or exceeds the bar for acceptance. They note the work is a novel, clinically-aligned, comprehensive solution moving toward an interpretable direction and is a good contribution to cardiac AI. While there are some remaining concerns, e.g., regarding latency, reviewers note that overall most concerns and clarifications were well-addressed by the rebuttal. Given the timely research topic, I believe the MIDL community will find this work of high interest.

---

### Decision · Program_Chairs · 2026-02-13

Accept (Poster)